# Breaking Sound Barriers: Exploring Tele-Audiology’s Impact on Hearing Healthcare

**DOI:** 10.3390/diagnostics14080856

**Published:** 2024-04-22

**Authors:** Mien-Jen Lin, Chin-Kuo Chen

**Affiliations:** 1School of Medicine, College of Medicine, Chang Gung University, Taoyuan 333323, Taiwan; j565243782@gmail.com; 2Department of Otolaryngology-Head and Neck Surgery, Chang Gung Memorial Hospital, Keelung 204201, Taiwan; 3School of Traditional Chinese Medicine, College of Medicine, Chang Gung University, Taoyuan 333323, Taiwan; 4Department of Otolaryngology-Head and Neck Surgery and Communication Enhancement Center, Chang Gung Memorial Hospital, Taoyuan 333423, Taiwan

**Keywords:** tele-audiology, telemedicine, telehealth, hearing screening, hearing aid, cochlear implant, hearing loss, smartphone, tablet

## Abstract

Hearing impairment is a global issue, affecting billions of people; however, there is a gap between the population affected by hearing loss and those able to access hearing healthcare. Tele-audiology, the application of telemedicine in audiology, serves as a new form of technology which aims to provide synchronous or asynchronous hearing healthcare. In this article, we reviewed some recent studies of tele-audiology-related topics to have a glimpse of the current development, associated challenges, and future advancement. Through the utilization of tele-audiology, patients can conveniently access hearing healthcare, and thus save travel costs and time. Recent studies indicate that remote hearing screening and intervention are non-inferior to the performance of traditional clinical pathways. However, despite its potential benefits, the implementation of tele-audiology faces numerous challenges, and audiologists have varying attitudes on this technology. Overcoming obstacles such as high infrastructure costs, limited reimbursement, and the lack of quality standards calls for concerted efforts to develop effective strategies. Ethical concerns, reimbursement, and patient privacy are all crucial aspects requiring in-depth discussion. Enhancing the education and training of students and healthcare workers, along with providing relevant resources, will contribute to a more efficient, systematic hearing healthcare. Future research will aim to develop integrated models with evidence-based protocols and incorporating AI to enhance the affordability and accessibility of hearing healthcare.

## 1. Introduction

Hearing impairment represents a significant global health challenge. According to the World Health Organization, approximately 1.5 billion individuals are affected, and is projected to rise to 2.5 billion by 2050 [1]. Out of this population, about one-third require hearing healthcare assistance, with 80% residing in low- and middle-income countries. It is also estimated that over 400 million people, including 34 million children, experience disabling hearing loss, impacting their health and quality of life. It currently ranks as the third most substantial contributor to years lived with disability. Hearing loss profoundly influences various aspects of life, including communication, social engagement, education, employment, overall well-being, and health [1,2,3,4]. There is also evidence suggesting a correlation between age-related hearing loss and cognition function as well as quality of life in older population [5,6]. However, despite the widespread occurrence of hearing loss, there is a gap between the number of people with hearing loss and those who can receive hearing healthcare [4,7]. A study reveals that around 400 million hearing-impaired individuals need hearing aids, yet most of them do not have one (83% in need are not using hearing aids), especially in the African region, and Eastern Mediterranean and Southeast Asia regions [4]. According to the National Institute on Deafness and Other Communication Disorders (NIDCD), for those who can benefit from using hearing aids, fewer than 30% of adults aged 70 or above and even fewer adults aged 20 to 69 (approximately 16%) have ever used them [8]. Therefore, it is seen that access to treatment for patients with hearing loss face some challenges.

Various elements influence an individual’s choice to seek hearing healthcare, and even those prepared to receive treatment may encounter obstacles on the way [9]. As an illustration, studies have indicated that patients residing in rural areas often face delays in accessing hearing healthcare services [10,11,12]. For the patients who manage to visit the clinic, travel cost, time, convenience, and disruption of family life can be a huge burden and cause resistance [13]. An unequal distribution of ear and hearing workforce is also a contributing factor, and it is related to the income levels of different regions [14]. Governments in many countries are also not aware of the importance of supporting and funding hearing healthcare, demonstrating a lack of attention and prioritization [3].

The conundrum has led hearing health experts to ponder: how can we minimize the gap of access inequality? Fortunately, a solution is introduced to the field: telemedicine. According to World Health Organization (WHO), telemedicine is defined as “the delivery of health care services, where distance is a critical factor, by all healthcare professionals using information and communication technologies for the exchange of valid information for diagnosis, treatment and prevention of disease and injuries, research and evaluation, and for the continuing education of health care providers” [15].

Starting from using the telegraph to send messages about wounded soldiers to medics during the civil war in the 1860s, telemedicine has evolved substantially with the assistance of the Internet and modern electronic devices such as computers, mobile devices, and smartphones [16]. Telemedicine facilitates remote patient–physician interactions, transcends geographical barriers, and offers a scalable and cost-effective solution to address healthcare disparities, particularly in underserved and rural communities [17]. Telemedicine has also found its application in audiology, referred to as tele-audiology, enabling the provision of remote services such as hearing screening, diagnostic testing, interventions, and rehabilitation, including tasks like hearing aid fine-tuning and cochlear implant adjustments [18,19].

During the COVID-19 pandemic, we witnessed significant disruptions in the global provision of healthcare services, with mandatory social distancing and lockdowns being the main hindrance. Under this challenging circumstance, hearing healthcare providers adopted tele-audiology to perform triage and offer healthcare services [20,21,22,23,24]. In this review article, we will discuss the question of how far tele-audiology healthcare have reached and can reach, with three main objectives including (i) the fields that tele-audiology has been applied to, (ii) challenges and issue of implementing tele-audiology/telemedicine, and (iii) the future advancement of tele-audiology/telemedicine and hearing healthcare.

Although there are different definitions of telemedicine and telehealth, in this article, we use these terms interchangeably to further discuss the potential of this technology.

## 2. Materials and Methods

The major body of the article would be discussing the three objectives mentioned in the introduction. This review article was completed between October 2023 to April 2024. The search and collection of articles were mainly carried out by the first author, with the assistance and examination by the second author. The initial search was performed on 7 October 2023 in three databases: PubMed, Google Scholar, and Scopus. The databases were selected for their extensive coverage, ensuring inclusion across a diverse array of fields. The research query included the following keywords: tele-audiology, teleaudiology, telemedicine, telehealth, tele-health, hearing loss, audiology, hearing aids, screening, testing, intervention, otoscopy, and audiometry. The database search was limited to recent 6 years (2018~2024). Eligible articles pertaining to our objective of discussion were included, while review articles, duplicates, and articles that are not related to our focus were ruled out. Articles not written in English were also excluded due to a lack of translation resources. The search query was conducted in each database based on specific requirements by employing Boolean logic. An initial search yielded 95 articles, with 23 articles from PubMed, 60 articles from Google Scholar, and 12 articles from Scopus. After removing duplicates and unrelated articles, we also added some relevant references manually. Final search results showed 63 references with 13 articles of hearing screening studies, 4 articles of diagnostic testing studies, 10 articles of hearing aids studies, 7 articles of cochlear implant studies, 4 articles of otoscopy studies, 6 articles of other interventions, 4 articles of barriers, 8 articles of major issues, and 7 articles of future advancement.

## 3. Implementation of Tele-Audiology

### 3.1. Hearing Screening

#### 3.1.1. Audiometry

Electronic devices such as computers, smartphones, and tablets are becoming popular tools for researchers to examine the utility of tele-audiology. Self-administered tests also offer accessibility advantages for remote locations and economically disadvantaged communities. Mealings et al. attempted to assess the agreement between self-administered and clinician-administered hearing tests in Aboriginal and Torres Strait Islander children [25]. It compared manual audiometry with two self-administered hearing testing apps. The apps did not generate 100% accuracy, but they could provide more efficiency on identifying children with hearing loss. Chu et al. adopted a smartphone-based approach for the hearing screening of school-age children [26]. They developed an iOS-based smartphone app with stratified hearing scales to assess the accuracy of a hearing test conducted on an iPhone. This study was characterized by the stratified hearing scale design which helped hearing health workers to better determine the hearing status of the tested ears, when compared with dichotomized results (pass or fail) from a pure-tone screening. The sensitivity and specificity both reached 100%, while the false-positive and false-negative results were 0%. With this app, hearing health workers were able to analyze the distribution of hearing scales and compare the screening results among classes and schools. Another study by Bowers et al. adopted a game-based hearing screening program which was used to examine the prevalence of hearing loss and spatial processing disorder (SPD) in a study sample of 1256 children aged 4–13 years [27]. By using a tablet-based app, this program was highlighted with enabling multiple children to be screened and more assessed auditory parameters at one time versus screening with pure-tone audiometry (PTA). As this program was carried out at eight schools, a further large-scale hearing screening would be feasible and useful in a timely manner, with less on-site expertise required.

While some authors like Chu et al. and Linkenheimer preferred bundle headphones over Bluetooth headphones for a better consistency of sound [26,28], Saunders et al. designed a study procedure involving automated audiometry and manual audiometry by using a wireless automated hearing testing system with over-ear headphones connected to a tablet via Bluetooth technology [29]. The study was carried out in a rural region in Nicaragua, with 3398 seven-to-nine-year-old school children recruited from the region. The automated audiometry pass rate was significantly lower than the manual audiometry pass rate (59% vs. 93%) due to many students having difficulty with automated audiometric testing. The results revealed an estimated 5.6% to 16.5% per 1000 children in rural regions having significant hearing loss. Saunders et al. confirmed that the ambient noise was the major challenge to hearing screening and identified two risk factors of sensorineural hearing loss (SNHL): maternal drug use and exposure to pesticide [29]. However, the analyzed data did not include the specific drug or pesticide which the children were exposed to.

Denga et al. employed a cross-sectional study of hearing screening with the tele-audiology-enabled KUDUwave audiometer in a low-resource community in Cape Town, South Africa [30]. The study validated that no significant difference of PTA results was observed between on-site screening and remote screening. Tele-audiology was proven to be a plausible solution for hearing health service delivery in rural or remote areas.

Community health workers (CHWs) can also contribute to hearing healthcare in many ways, especially with telemedicine technologies. In two similar studies, Eksteen et al. and Manus et al. demonstrated that CHWs using mobile health technology (mHealth) successfully screened eligible preschool children for hearing and vision issues in South Africa, with a low cost of $5.63 and $6.67 per child [31,32]. Another study by Dawood et al. revealed that there was no significant difference in smartphone screening audiometry outcome between CHWs and school health nurses except for testing duration, with CHWs spending more time to conduct the test on average (69 s vs. 56 s) [33]. The authors mentioned the difference in testing duration was statistically significant but clinically insignificant. Both studies concluded that longer test durations and excessive background noise were associated with hearing test failures.

Gos et al. estimated the prevalence of hearing loss in children of a rural region in Poland by performing PTA testing on children in quiet classrooms and using the Internet to transmit PTA results’ data from remote portable computers to a central computer [34]. The estimated prevalence of hearing loss was 7%, which was similar to some previous studies mentioned in the article. Meanwhile, younger children were found to be more likely to suffer from hearing loss than older ones, which was attributed to the higher incidence of otitis media following upper respiratory tract infection and eustachian tube dysfunction. For young children in rural region, the early detection and treatment of hearing problems would be beneficial to their education, which is the reason that telemedicine can be adopted to fill the gap of unequal access to hearing health services.

S. M. Govender and M. Mars designed a comparative within-subject study and examined the efficacy of telehealth-based hearing screening and diagnostic testing in a rural primary school in South Africa [35]. With a trained facilitator performing the otoscopic examination and assisting in the automated pure-tone screening, the hearing results of participating children would be evaluated by a qualified audiologist. The audiologist then conducted on-site diagnostic automated audiometry. The screening results showed a specificity of 100% but a low sensitivity of 65.2%, which was attributed to device-related factors, child-related variables, and the missed detection of mild hearing loss. However, this study demonstrated that a trained facilitator could perform video-otoscopy and pure-tone screening semiautomatically. Further on-site diagnostic testing would not be required, with the data of subjects directly transmitted to the audiologist via the Internet and analyzed by the audiologist synchronously or asynchronously. A comprehensive set of audiologic tests is possible within a telehealth model in the future.

#### 3.1.2. Distortion Product Otoacoustic Emissions (DPOAEs)

In a study performed in Ghana, Ameyaw et al. compared the results and duration of the newborn hearing screening test (DPOAE test in this study) conducted in a telehealth method group and conventional on-site method group [36]. The telehealth testing process involved a facilitator assisting in DPOAE probe insertion and computer application at the remote site, and an audiologist conducting real-time DPOAE screening tests online. For the conventional method group, face-to-face DPOAE hearing screening was performed as the control for the telehealth group. After analysis, there was a strong correlation between the two groups in the DPOAE test results. The telehealth method group was 2.6 s slower than the conventional on-site method group, but the difference was not statistically significant. The delay in the telehealth group might result from the delay of the “command-to-response” time between the host computer and the remote one. The author suggested that, with the help of the telehealth method, hospitals in different regions could be linked to the major hospital as a hub in Ghana, where the majority of healthcare resources were located. However, despite tele-audiology offering a valuable opportunity for minimizing the regional inequity of hearing healthcare, its usefulness is currently hampered by the lack of readily available, appropriate equipment in remote areas in the world.

#### 3.1.3. Speech in Noise Test

Paglialonga et al. developed an automated speech-in-noise test which included “vowel–consonant–vowel” sounds in a discrimination test of multiple-choice questions [37]. The proposed test adopted a one-up/three-down staircase method (the signal-to-noise ratio was increased after one incorrect response and decreased after three correct responses) to provide an accurate estimation of the speech reception threshold (SRT). The outcomes showed an accurate estimation of the SRT and reliable test–retest repeatability. However, a larger study sample size is necessary for a more thorough and comprehensive analysis on accuracy and reliability in the future. Ratanjee-Vanmali et al. proposed a hybrid model of hearing healthcare, which was a combination of online and face-to-face modes of synchronous and asynchronous communication for patients and healthcare providers [38]. The study recruited 462 individuals out of 1852 website visitors to complete the online digit-in-noise (DIN) test, with 59% (271/462) failing the test and 11% submitting their contact details. The online DIN test is a triple-digit hearing screening test. A strong association between age and the DIN test results was confirmed. People who passed the DIN test were significantly younger than those who failed the test (*p* < 0.001). Eventually, five individuals completed a thorough face-to-face hearing evaluation and hearing aid trial. These individuals were all aware of their hearing problems before taking the test and were ready to seek further hearing healthcare. This study demonstrated the promise of integrating the hearing screening, evaluation, hearing solution, and continued support by using the hybrid models of hearing healthcare.

### 3.2. Diagnostic Testing

Tele-audiology has proven to be a valuable tool for otologists in the diagnosis of hearing disorders. For example, Habib et al. evaluated the use of tele-audiology for diagnosing otitis media in indigenous children living in rural and remote areas of Queensland, Australia [39]. Their findings revealed that diagnostic accuracy improved progressively with the availability of more clinical data: 65% accuracy with otoscopic images alone (Tier A), 77% with otoscopic images plus tympanometry and category of hearing loss (Tier B), and 85% with additional data including static compliance, canal volume, pure-tone audiometry, and nurse impressions (Tier C). Inter-rater agreement also improved with increasing clinical data, and the largest improvement in classification accuracy was observed when comparing Tier A to Tier C, demonstrating the value of incorporating additional clinical information in tele-audiology assessments for ear disease diagnosis.

V. Ramkumar and M. Krumm conducted a pilot study which provided a viable tele-audiology model for remote diagnostic testing, with one audiologist serving as the on-site facilitator, and the other audiologist conducting all the hearing tests using remote control of the audiology equipment and using video-conferencing to interact with the subject and the facilitator [40]. The diagnostic test battery included a series of video-otoscopy, tympanometry, DPOAE, and ABR test. The study sample consisted of 10 typically developing children (pilot norm) and 7 children with disabilities, and, therefore, co-facilitators were required for children with disabilities to assist in the facilitator in terms of each child’s unique communication methods. All children completed video-otoscopy and tympanometry, and most of the children completed DPOAE testing. This study served as the cornerstone of a more comprehensive audiologic diagnostic testing following studies conducted by researchers such as S. M. Govender and M. Mars [35]. Ramkumar et al. also conducted a study where two diagnostic ABR testing models were employed: one via telemedicine and the other through traditional in-person testing at a tertiary care hospital [41]. The program achieved a low refer rate (0.8%) and a high follow-up rate (80–97%) for second screenings, with tele-ABR follow-up showing an 11% improvement in adherence compared to in-person ABR at the hospital, highlighting the potential of telemedicine in enhancing follow-up care in resource-constrained settings.

For emergent cases, tele-audiology can provide automated diagnostic testing which is reliable and useful for identifying patients in need of urgent care but are away from medical resources. E. S. Linkenheimer reported a case of a patient with idiopathic sudden sensorineural hearing loss using virtual hearing testing on mobile devices at home to evaluate the efficacy of the treatment during the COVID-19 pandemic [28]. The patient experienced a subjective improvement, which corresponds to the virtual audiogram results. The virtual audiogram can be further used to evaluate asymmetry of hearing, compare the present and previous audiogram, and triage patients to assess which patients require in-person clinic visits via tele-audiology. In a prospective cohort study, Shilo et al. explore the feasibility of a telemedicine model for assessing new-onset unilateral sudden hearing loss (SHL) during the COVID-19 pandemic [42]. The telemedicine model involved a Weber test and smartphone-based audiometry, demonstrating a high sensitivity of 100% and specificity of 73% in discriminating patients with sudden sensorineural hearing loss (SSNHL) from those without. The telemedicine model proved to be a valid and reliable tool for identifying patients in need of urgent care for SSNHL, especially when access to in-person medical resources is limited during the pandemic.

### 3.3. Hearing Aids

There is plenty of room for improvement required to elevate the prevalence of hearing aid use. Orji et al. extracted the data of hearing aid use from large surveys and a systematic review to estimate the global and regional needs and unmet needs (those in need of a hearing aid but do not use one) for hearing aids [4]. The results indicated that 401.4 million hearing-impaired individuals in the world need hearing aids, while 83% in need do not use a hearing aid. Moreover, if every individual in need of a hearing aid would use one, the burden of the disease would be reduced from 25 million years lived with disability (YLDs) to 10.3% YLDs, which would stand for a decrease of 59%. The prevalence of hearing aid use is also low in the US, especially among ethnic minorities and individuals with a lower income. Coco et al. examined the feasibility of a tele-audiology approach to deliver hearing aid service to rural, low-income Hispanic/Latino adults with bilateral hearing loss with the help of local CHWs (experimental group) and trained university students (control group) as patient-site facilitators [43]. Both the CHW group and the student group showed improved hearing outcome. CHWs played a pivotal role in the study, such as reaching and maintaining trust among the involved participants. The potential of tele-audiology combined with CHW support can be huge in terms of expanding access to hearing care.

In a randomized controlled trial conducted by Convery et al., with thirty adult patients involved, hearing aid performance as well as the benefit, satisfaction, and daily usage were compared between a hearing aid app group and face-to-face appointment group [44]. The app was found to be user-friendly, and no significant difference in hearing aid outcomes was observed between remote and in-person groups. However, in-person care was still necessary regarding many requests other than fine-tuning from the patients. Tao et al. conducted a single-blinded crossover randomized controlled trial in which participants received both face-to-face and tele-audiology follow-up consultations for hearing aid fitting [45]. Assessments were conducted before and after the consultations, using various outcome measurement tools. The study found that tele-audiology hearing aid follow-up consultations are equally effective as standard face-to-face consultations, with both modes significantly improving communication, fitting, and quality of life; however, satisfaction may be slightly lower with tele-audiology due to technical or human-related issues. Venail et al. compared the face-to-face and remote programming of hearing aids in experienced users and found that both methods resulted in similar speech perception outcomes and real-ear insertion gains [46]. The study also reported no significant differences in self-perceived hearing benefits, indicating that remote programming is as effective as in-person programming without requiring additional time spent on patient care. Moreover, Lau et al. found that remote consultations and assessments for patients awaiting consideration of bone conduction hearing devices (BCHDs) during the COVID-19 pandemic were effective, with patients expressing satisfaction with the remote assessment, suggesting the potential for continued use in reducing hospital visits beyond the pandemic [47].

Evans et al. reported three cases of tele-audiology application on hearing aid evaluation, hearing aid validation, and auditory processing evaluation, respectively [48]. All three cases were successfully completed with tele-audiology methods, which showed the efficiency of tele-audiology practice and reduced hospital trips during the COVID-19 pandemic. David P. Jedlicka reported a case of same-day hearing aid evaluation and fitting, along with a subsequent telehealth follow-up [49]. The result of a successful fitting was encouraging with the patient’s self-perceived improvement and datalogging showing the full-time use of the hearing aids. The telehealth follow-up turned out to be a multidisciplinary and patient-centered approach for not only the outcome measures of the hearing aids but also the treatment of the patient’s mental health issue.

Furthermore, using multimedia as an educational approach can improve patients’ self-efficacy, which mainly consists of self-management and knowledge regarding hearing aids. Rachel Gomez and Melanie Ferguson discovered an improved self-efficacy for first-time hearing aid users via reusable learning objects (RLOs), such as video clips, illustrations, animations, photos, and sounds provided in the study [18]. The study also confirmed that patients’ adherence to RLOs were higher than their adherence to educational booklets. In a randomized controlled trial performed by Muñoz et al., 82 parents of children (birth to 42 months) who used hearing aids were included in the hearing aid education program, and half of them were assigned to the eHealth group while the other half were assigned to the treatment-as-usual group [50]. The eHealth program contained eight videos of tutorials and weekly phone calls to assess the participants’ experience with the program and the progress they had made. The eHealth group showed improvements in knowledge, parent perceptions, confidence, and monitoring compared to the control group, with significant differences in knowledge and confidence at the end of the study. Muñoz et al. recommended further research is needed, including a larger, more diverse demographic sample and more dynamic support from the healthcare providers [50].

In brief, hearing aids are underutilized globally, with most needing them not having them. Tele-audiology, providing hearing aid services remotely, shows promise in increasing access. Studies suggest tele-audiology consultations and programming are just as effective as in-person visits, with potential cost benefits. Research is ongoing to improve access and empower patients with hearing loss through tele-audiology and educational approaches. However, it is still necessary for the patients to have one face-to-face contact with the hearing aid providers to ensure the process of using hearing aids is comfortable and adequate for the patients.

#### Auditory Brainstem Response (ABR)

One study by Ramkumar et al. implemented a community-based pediatric hearing screening program in remote rural areas using trained village health workers (VHWs) [51]. Two diagnostic ABR testing models were employed: one via telemedicine and the other through traditional in-person testing at a tertiary care hospital. The program achieved a low refer rate (0.8%) and a high follow-up rate (80–97%) for second screenings, with tele-ABR follow-up showing an 11% improvement in adherence compared to in-person ABR at the hospital, highlighting the potential of telemedicine in enhancing follow-up care in resource-constrained settings [51]. In addition, Hatton et al. [52] revealed that the telehealth-enabled auditory-brainstem-response (TH-ABR) initiative, as part of the British Columbia Early Hearing Program (BCEHP), proved to be cost-effective and sustainable, covering its costs after conducting 102 TH-ABR assessments and successfully identifying 50 infants with hearing loss. The accuracy and efficiency of TH-ABR were on par with in-person assessments, and feedback from parents via surveys showed a strong level of satisfaction. While tele-ABR can help expand the coverage of hearing healthcare, one should not overlook the fact that remote ABR measurements still require trained professionals to perform, which differs from tele-audiometry.

### 3.4. Cochlear Implants

The entire treatment process of cochlear implants (CIs) can be lengthy and cumbersome for both patients and healthcare providers. With the assistance of telemedicine, the inefficiency may be improved in multiple dimensions. In a prospective pilot study, the feasibility of examining cochlear implant candidacy was tested by Fletcher et al. through a telemedicine video-conferencing approach, showing similar results in comparison to traditional in-person evaluations [53]. In another prospective pilot study, Lohmann et al. examined the feasibility of the automated remote intraoperative cochlear implant reverse telemetry testing by using intraoperative Remote Assistant (Cochlear Nucleus CR120) [54]. Unlike some smartphone-based tele-audiology which can be operated by patients themselves, the intraoperative Remote Assistant can only be performed by trained operating room staff. Automated remote reverse telemetry testing was completed in all cases, with the results of minimal discrepancies between manual testing and remote testing on the impedance and Electrically Evoked Compound Action Potential (ECAP) response. The CR120 remote system required less testing time compared to standard manual testing.

Benjamin Boss examined a case of troubleshooting a cochlear implant processor by offering a virtual appointment to the patient, a 92-year-old male with a cochlear implant in his left ear [55]. During the appointment, the patient reported issues such as insufficient battery life and a pairing problem of the processor with his iPhone. With the capability of tele-audiology, the issues mentioned were able to be identified and resolved in later actions. Cullington et al. made a comparison between the standard clinical care and remote care of cochlear implant patients, demonstrating that patients receiving remote care had a greater improvement on the Triple Digit Test and the patient activation measure questionnaire, whereas patients receiving standard clinical care reported a worse performance in the speech, spatial, and qualities questionnaire [13]. There was no significant difference in the quality of life of both groups. Nevertheless, Cullington et al. recommended that a further larger-scale study is needed, and the training effect of the remote care group and limitations of the questionnaire require more careful evaluation [13].

Two studies showed a similar interest in discussing the safety and efficacy of remote CI programming. Schepers et al. assessed the process of remote CI programming in experienced users including children and adults and compared it with a local fitting [56]. The outcomes showed that remote CI programming is a safe, feasible, and effective approach with no significant difference found in impedance field telemetry, maximum comfortable levels, threshold levels, audiometry, and speech understanding. Slager et al. also investigated the safety and efficacy of remote CI programming, with the results of no device and/or procedure-related adverse events noted, and a non-inferior performance of remote programming compared to the in-office programming [57]. However, it is reminded that remote programming is not suitable for all cochlear implant recipients, especially those requiring more direct attention, extensive counseling, or complex troubleshooting. Both studies found that remote programming yields non-inferior speech perception outcomes and high levels of satisfaction among patients, audiologists, and medical professionals with remote fitting. Nonetheless, both studies mentioned issues related to reimbursement, state licensure, and the need for in-person visits for certain cases.

Nassiri et al. introduced the Complete Cochlear Implant Care (CCIC) model, which streamlines cochlear implant care delivery to a single on-site visit, including same-day on-site CI evaluation, same-day surgery, same-or-next-day activation, and postoperative programming for 12 months [58]. The CCIC model primarily utilizes telemedicine and electronic educational materials to perform preoperative education, telemedicine appointments, and postoperative remote programming [58]. The collected data from the CCIC model clinical trial indicated high patient satisfaction and noninferior clinical outcomes at one month postoperatively. While challenges were noted, such as adapting to new clinical workflows and hardware requirements for remote programming, the CCIC model has the potential to significantly improve hearing healthcare delivery.

### 3.5. Cognitive Behavioural Therapy for Tinnitus via the Internet (iCBT) and Progressive Tinnitus Management (PTM) by Telephone (Tele-PTM)

iCBT is an Internet-based therapy that is delivered through a computer or mobile device and follows the principles of Cognitive Behavioral Therapy (CBT). There are many different modules with the content of CBT, such as applied relaxation, thought analysis, cognitive restructuring, imagery, and exposure techniques. Beukes et al. tested the efficacy of reducing tinnitus distress and comorbidities with audiologist-guided iCBT in a randomized controlled trial consisting of 146 participants from the United Kingdom, who were randomly assigned to the iCBT group and the weekly-monitoring control group [59]. Tinnitus distress was measured using the Tinnitus Functional Index (TFI), and significant improvements were observed in the experimental group compared to the control group. Symptoms of various tinnitus comorbidities, including insomnia, depression, hyperacusis, and cognitive failures, were significantly reduced through iCBT. The study suggests that the medium effect size found in the present trial aligns with values from previous iCBT trials for tinnitus. A similar study conducted by Beukes et al. in Texas, United States also showed that iCBT had a greater effect on reducing tinnitus distress and greater reduction in negative tinnitus cognition and insomnia than weekly monitoring [60]. However, this study experienced low completion rates for post-treatment questionnaires compared to similar iCBT studies for tinnitus. Only 6% withdrew, but many enrolled participants never logged into the intervention website. Compliance was observed to be low, which could be attributed to an additional assessment time point for the control group and the free iCBT intervention, which might have led to the undervaluation of the treatment, particularly in a country where many people pay for healthcare. The authors suggested that a process evaluation could help identify factors influencing retention and engagement rates. Continued public involvement in planning and implementing future research trials is deemed vital for gaining insights into the factors important to participants.

Progressive Tinnitus Management, or PTM, is a stepped-care program that involves co-ordinated care between audiology and behavioral health. Tele-PTM is the connected-care (telehealth) version of PTM. Henry et al. found positive results for Tele-PTM for tinnitus patients with or without traumatic brain injury (TBI) compared to individuals on a 6-month waiting list for tinnitus services [61]. The improvements included a reduction in tinnitus functional effects, increased self-efficacy, and smaller-scale reductions in anxiety and depression. This study surpassed previous studies, with a greater average reduction in the Tinnitus Handicap Inventory (THI) score at 24 weeks post-baseline and a larger effect size for Tele-PTM. The lack of face-to-face contact in the telephone delivery of PTM did not hinder positive outcomes, with clinicians establishing a rapport and participants reporting positive experiences. Participants with moderate/severe TBI, who tended to fatigue more quickly and had more difficulty remembering appointments, would benefit from the flexibility and extra support identified as essential during the intervention. Both iCBT and tele-PTM require healthcare professionals such as audiologists, psychologists, and clinicians to provide remote guidance and support to the patients. However, the optimal delivery methods for both models are still being investigated. 

### 3.6. Otoscopy

Tele-otoscopy, a novel examination technique powered by telemedicine, has been studied and developing during the pandemic. Various otoscopes are designed to allow non-specialists, including patients, their parents, or primary healthcare providers to capture images of the eardrum and middle ear, enabling remote evaluation by ear, nose, and throat specialists (otolaryngologists). However, there are mixed results of its efficacy. A prospective study by Meng et al. demonstrated the effectiveness of smartphone otoscope telemedicine in a rural medical consortium in East China during the COVID-19 era [62]. The procedure involved primary healthcare providers using a smartphone otoscope to perform an examination, sending videos and photos of the patients’ ears to the otolaryngologists via WeChat group. The results showed that this telemedicine approach improved the ability of rural primary healthcare providers to diagnose and treat ear diseases, was well-received by patients, and was considered helpful by the primary healthcare providers, ultimately saving time and costs for patients.

Erkkola et al. examined the feasibility of a smartphone otoscopy performed by parents for the diagnosis of otitis media [63]. The study results demonstrated that parents could perform smartphone otoscopy, and physicians were able to detect or exclude Acute Otitis Media (AOM) in most parent-performed videos (87%). However, parent-performed smartphone otoscopy had only a moderate accuracy (40% of the videos of diagnostic value) in determining specific diagnoses (healthy ear, Otitis Media with Effusion, and AOM). While parents showed the ability to perform smartphone otoscopy, teaching them how to use one was deemed necessary for optimal performance. Parental experiences with the smartphone otoscope at home were generally positive. The study suggests the need for product improvements in smartphone otoscope technology, while further research is required to assess the clinical usefulness of parent-performed smartphone otoscopy and explore the possibility of automatic tympanic membrane image analysis systems. 

In another study conducted by Shah et al., CellScope iPhone Otoscope is used for comparing video exams recorded by parents to those recorded by an otolaryngologist [64]. The results showed a low agreement between a diagnosis based on parent-recorded videos and pneumatic otoscopy, while a diagnosis based on physician-recorded videos had a higher agreement with pneumatic otoscopy. Objective landmarks of the tympanic membrane on the videos also showed only a slight agreement between physician and parent recordings, indicating that, currently, parent-recorded images may not be suitable for diagnosis in tele-otoscopy. Finally, Dai et al. assessed the outcomes and satisfaction of ear patients participating in smartphone otoscope telemedicine via WeChat [65]. The results showed that this telemedicine approach effectively reduced outpatient visits, minimized the risk of cross-infection, enhanced telemedicine accuracy, and resulted in high patient satisfaction, suggesting its clinical applicability during the COVID-19 era. The major cause of different outcomes of the two studies mentioned may stem from several factors. In the former study, the CellScope device and instructions were provided to parents originally, along with the training video. However, parents received no assistance and guidance from the research staff. Moreover, parents failed to visualize the tympanic membrane with the device, often visualizing only the external auditory canal or the cerumen. On the other hand, in the latter study, patients received quick training from the otolaryngologists, and they were added to the WeChat community to communicate with their otolaryngologists online in text and voice message, and they could also upload their results of the periodic ear self-examination and pictures via WeChat. This demonstrates that tele-otoscopy technology may yield inaccurate results without healthcare professional assistance or guidance.

### 3.7. Referral

In a cluster randomized controlled trial, Emmett et al. evaluated the impact of telemedicine specialty referral on follow-up times after a school hearing screening in 15 rural Alaska Native communities, with an estimated sample size of 1500 children [66]. Specialty referral including telemedicine follow-up appointments for children required referral. If further otolaryngology consultation was needed, audiology providers would request a telemedicine consultation for surgical and medical management. Meanwhile, standard primary care referral included referral letters sent from schools to children requiring referral, requesting their parents to bring them to the village health clinic for evaluation. Later, children who required treatment would have three options: wait for an audiology field clinic (held every 3~4 months); a telemedicine audiology consultation, or a referral to a primary care provider. The outcome revealed that telemedicine specialty referral significantly reduced the time to follow-up compared to standard primary care referral, with the proportion of children receiving follow-up more than doubled in the telemedicine group. This study highlights the potential of telemedicine to enhance access to hearing specialty care for rural children in preventive school-based services. Similar studies by Robler et al., also conducted in 15 rural Alaskan communities, demonstrated that children receiving follow-up care after the hearing screening were significantly more represented in the telemedicine specialty referral group than the standard primary care referral group [67,68]. Upon integrating the quantitative and qualitative data from intervention communities, four of the six studied factors showed an association between the telemedicine specialty referral pathway: clinic capacity, personnel ownership and engagement, communication, and awareness of the need for follow-up. Robler et al. mentioned limitations in both studies of a small sample size, which could not be representative of the region’s population [67,68]. There are ongoing large-scale implementation trials taking place, which will adapt the intervention based on trial results and community feedback to provide a comprehensive understanding of the telemedicine intervention.

## 4. Barriers and Challenges

The implementation of telemedicine relies on several key factors, such as the technological infrastructure, adaptation of clinical workflows, and provision of training for both healthcare providers and patients, which are all essential for facilitating the remote care delivery [69,70,71,72]. However, there are still some barriers and challenges ahead of the way (Table 1). Alami et al. conducted a multi-project evaluation synthesis, listing some technological hurdles in telemedicine such as extensive storage requirements, emerging types of errors, overdiagnosis with a high image resolution, managing multiple software components, limited bandwidth in remote areas, and technological malfunctions affecting the reputation of healthcare providers and organizations [70]. The lack of computer and technology literacy could also be a significant obstacle.

In a study of surveying the perceptions of audiologists regarding tele-audiology in South Africa, Bhamjee et al. identified some perceived barriers after a thematic analysis [71]. The consensus among audiologists was that remote hearing healthcare is not accessible to all populations, with over 90% expressing this viewpoint. This perception was closely trailed by concerns regarding limited equipment, technological infrastructure, and the reliability of the Internet. Some audiologists reported a preference for face-to-face consultations and emphasized that the quality of service cannot be replaced by telehealth. There was also a lack of policy, protocols, and guidelines for the use of telehealth, along with insufficient knowledge and training for healthcare workers. Financial constraints were highlighted, as many patients struggle to afford the resources required for accessing telehealth services.

Barriers to tele-audiology were also identified by Parmar et al., including hearing healthcare services that cannot be performed remotely, the lack of quality standards or clinical guidance, and concerns about the evidence base for tele-audiology [72]. Patient-related barriers, such as access to and confidence in information communication technology, were highlighted. There were also concerns about confidentiality and patient data protection. Suggestions included considering tele-audiology provision on a case-by-case basis, developing practice advice for service provision, and strengthening the evidence base for tele-audiology. To sum up, these identified barriers not only demonstrate that much more effort needs to be put in to improve tele-audiology services, but also lead to a fundamental transformation of hearing healthcare.

## 5. Major Issues

### 5.1. Ethical Concerns

The World Health Organization (WHO) highlighted that telemedicine raises unique ethical concerns, including the preservation of patient confidentiality and privacy, as well as the protection of information system integrity [70,73]. It is crucial to ensure the protection of patient information while permitting the sharing of information that is essential to the research of health promotion at the same time [73]. Therefore, a more comprehensive discussion of topics involving pending ethical problems is required, such as the authorization of access to online measured data and the best practices for implementing data security measures during data transfer. There is the need for updated guidelines for the ethical use of telemedicine by clinicians and organizations. This is vital because clear and up-to-date guidelines can ensure that healthcare providers and organizations adhere to ethical principles while delivering care remotely, ultimately promoting the well-being and safety of patients [74]. Telemedicine projects raise concerns about shared responsibilities among various stakeholders, including potential challenges in assigning blame in cases of patient harm, and legal responsibility for patient data. Furthermore, the use of diverse technology components from different providers in telemedicine systems complicates accountability issues, especially with subcontractors and offshore data storage providers [70].

### 5.2. Cost and Reimbursement

Every institution may have different policies and regulations regarding cost management. For example, at Cincinnati Children’s Hospital Medical Center (CCHMC), individual departments are not charged for telehealth-related services due to the policy of prioritizing telehealth service provision in CCHMC [75]. The setup of telehealth provision costs a great amount of money and time, with the most expensive one being the video-conferencing system, which costs hundreds to thousands of dollars depending on which system is used. Telehealth-related equipment (such as webcams and headphones) and service licenses are also involved. For instance, costs for video conferencing licenses range from $200.00 to $400.00 for the first year, with an annual fee of $30.00 thereafter. From the perspective of clinic owners and managers, tele-audiology consultations may be as or more costly (financial and time costs) than in-person services [76]. The extra expenses related to staffing, time, and financial investments need to be considered for establishing and sustaining tele-audiology business models.

In the United States, the insurance payment for telehealth services varies among state, federal, and commercial payers, and, for the most part, the payment is limited [19,75]. In some of the states, there is a lack of state laws requiring private insurers to pay for telehealth-related services. Moreover, audiologists are not acknowledged as telehealth service providers by Medicaid and Medicare, leading to a lack of reimbursement for audiology services delivered through telehealth. Medicaid and Medicare do cover services from recognized telehealth providers such as physicians, physician assistants, psychologists, and social workers, but limitations often exist regarding the patient’s location during service delivery. In Australia, the first seven months of the government allows the reimbursement of hearing services delivered via tele-audiology modes [76]. Financial reimbursement is the major driving force for clinic owners/managers to adopt tele-audiology, so the funding frameworks need to be clear and adequate. With cost and reimbursement issues taken into account, more specific regulations and laws need to be established and implemented for the development of telehealth services in the future.

### 5.3. Licensure

The absence of an international licensure agreement presently restricts the widespread adoption of tele-audiology across countries and regions. In the US, licensure can be problematic due to different legal and regulatory requirements. According to American Speech–Language–Hearing Association (ASHA) guidelines, tele-practitioners must hold licenses in both the service-providing state and the state where clients are located during sessions. Specific licensing requirements may differ for civilian employees, so confirmation is necessary [19]. Furthermore, licensure often emphasizes the physical location of the provider rather than the patient’s location, which restricts access to care, especially in rural areas with limited specialists. In other words, the licensure system in tele-audiology is still a complete mayhem. Currently, there are some established laws and bill proposals on the way, such as the Audiology & Speech–Language Pathology Interstate Compact (ASLP-IC), which has been established to allow professionals to practice in multiple states without additional licenses. State legislation regarding tele-audiology licensure has also been proposed, supported with or without amendments, and established [77]. 

To solve the problem of the discrepancy between different states or regions, it is crucial to establish national licensure standards or a centralized system for recognizing state licenses, which would streamline practice and improve patient access [78]. Licensure regulations should consider the patient’s location as the primary factor for service delivery, not the provider’s. Licensure frameworks should potentially consider the specific technologies used in tele-audiology and any associated security measures. In terms of solutions, organizations like the American Academy of Audiology can advocate for model licensure laws that different states can adopt, which promotes consistency. Developing interstate compacts for audiology licensure, similar to existing ones for other professions, could facilitate practice across state lines. By addressing these issues, the future of tele-audiology licensure can be more efficient, accessible, and patient-centered.

### 5.4. Security and Privacy

Information Technology (IT) systems face constant and diverse threats, including attacks on computers, devices, networks, applications, and the people and processes involved [79]. Tele-audiology, like other IT systems, deals with potential vulnerabilities at the device and business levels. Attacks on medical devices can have severe consequences, including life-threatening situations, economic losses, reputation damage, litigation, and privacy breaches. We provide an example of the HeLe Newborn Hearing Screening (NHS) Tele-Audiology Systems in the Philippines [80]. The systems were set up using containerization technologies and secured by layers of protection such as reverse proxy, transport layer security, secure shell, and host-based firewalls. With these standard network procedures carried out, the protection of confidential information of both patients and health institutions can be ensured. It is also recommended that researchers follow current national policy and administrative measures regarding patient privacy issue.

## 6. The Future of Tele-Audiology

Despite the barriers and challenges ahead, tele-audiology still has great potential in various ways. The importance of tele-audiology in the field of hearing healthcare should not be underestimated, and, therefore, in the future, including tele-audiology in education program for audiology students seems necessary, which can enhance theoretical knowledge and practical competency among students [81]. Training healthcare providers with computer-based courses of tele-audiology is a feasible way to facilitate the implementation of tele-audiology in rural regions [82]. To accomplish this, the IT infrastructure must be established and further updated to minimize the gaps existing in low resource settings.

Audiologists’ opinions toward tele-audiology vary greatly in many dimensions. In an Internet-based questionnaire survey performed among 108 audiologists, MR NN and Seethapathy examined the domains of the knowledge, attitude, and practice of tele-audiology among Indian audiologists [83]. The results showed that a high score was noted in the knowledge domain, but there was a mixed attitude toward tele-practice and whether tele-practice would replace face-to-face services in the future. Furthermore, the practice of tele-audiology is quite limited currently, with many problems mentioned pending to be resolved. Evidence-based practice is required for a more comprehensive and systematic telehealth model, and, thus, encourages more tele-audiology practice among audiologists in the future.

In clinics or pharmacies, it is likely to see technologies like free-standing kiosks or digital notepads or tablets with noise-cancelling headphones, which can provide a rapid hearing screening test with immediate, reliable results [84]. For audiologists, integrating in a partnership with allied health professionals, such as optometrists, ophthalmologists, podiatrists, and pharmacists, may enhance and extend hearing health services. Patients are also able to gain access of multiple healthcare services with less travel costs and time.

In a project that is underway, Weinreb et al. manage to adapt the technology used by Intelehealth, a nonprofit startup in India, to equip the frontline healthcare workers with open-source digital assistant and telemedicine platform [85]. With age-related hearing loss (ARHL) being the largest modifiable risk factor for dementia, yet with only 15% of ARHL patients using hearing aids on a regular basis, Weinreb et al. attempted to establish an integrated model of decision support and a telemedicine platform for the targeted intervention of maximizing hearing aid access and acceptability among older patients [85].

In a world where technology never stop its evolution, artificial intelligence (AI) has the potential to streamline remote care delivery and could expedite the global acceptance of tele-audiology services at cost-effective rates. Thus far, AI has already been used in several aspects of hearing healthcare, such as personalized recommendation engines finding the most suitable hearing aids based on patients’ needs and lifestyle, and cochlear implant programming which yields an equivalent performance compared to experienced clinician standard programming [86,87]. The future application of the combined power of AI and telemedicine technology will undoubtedly facilitate the development of hearing healthcare and provide novel solutions for hearing-related issues.

## 7. Conclusions

With the assistance of tele-audiology, patients have easier access to hearing healthcare with reduced travel costs and time. They also benefit from automated service and video-conferencing with healthcare professionals during emergent times like the COVID-19 pandemic. Recent studies showed that remote hearing screening and intervention yield the equivalent performance of healthcare professionals, and sometimes even outscore the traditional standard clinical pathway. Nevertheless, there are still piles of barriers and challenges to implementing tele-audiology, while audiologists hold different opinions toward this technology. Major issues such as the high costs of infrastructure, shortage of reimbursement, and data confidentiality require much more effort to come up with better solutions. Educating and training students or healthcare workers with related materials will improve the quality and efficiency of care. Future studies will provide integrated models with evidence-based protocols and practices, and continue to promote a multidisciplinary partnership and fusion with AI to deliver affordable and accessible hearing healthcare.

## Figures and Tables

**Table 1 diagnostics-14-00856-t001:** A comparison of major findings of identified barriers.

Findings	Bhamjee et al. [71]	Parmar et al. [72]	Alami et al. [70]
Similar barriers	Hearing healthcare not accessible to all Inadequate technological infrastructure Lack of policy and protocols Limited equipment Uncertain reliability of the Internet Preference for face-to-face consultations Quality of service cannot be replaced by telehealth Insufficient knowledge and training for health workers Financial affordability for patients	Lack of access and confidence for patients Healthcare cannot be performed remotely Lack of quality standards or clinical guidance Limited equipment for remote assessing Uncomfortable with remote consultations Confidentiality and patient data protection	Inequalities in access technology and digital literacy Unintegrated software or technologies Tension between standardization and local practice Overconfidence in technology; extensive storage requirements Limited Internet bandwidth Concerns of depersonalization using telemedicine Uncertain cost-effectiveness of telehealth Concerns about data use by governments or commercial purpose
Other findings		Concerns of evidence base for tele-audiology	Overdiagnosis with high image resolution Clinical data sent by error or received without validation Change in existing clinical workflow Concerns about clinicians leaving small hospitals Having multiple stakeholders leads to a dilution of responsibilities

## Data Availability

Data sharing is not applicable.

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
