# Peer review of "Breaking Sound Barriers: Exploring Tele-Audiology’s Impact on Hearing Healthcare"

_diagnostics, 2024, doi:10.3390/diagnostics14080856_

Round 1

Reviewer 1 Report

Comments and Suggestions for Authors

Line 24 please, add some words in abstract/paragraph to ethical aspects and safety c.q. privacy issues as they are prominent and important in tele-audiology.

Line/paragraph 187 DPOAE’s: please, add some words to this paragraph about lack of remote possibilities to use DPOAE measurements for hearing screening purposes, due to the unavailability of adequate DPOAE apparatus in remote situations everywhere, despite what has been done in Ghana.

Line/paragraph 198 SiN Test: please, add some words to the study https://pubmed.ncbi.nlm.nih.gov/30289050/ about using DIN-test in tele-audiology, e.g. because I think this possibility has a worldwide spread today.

Line/paragraph 250 Hearing aids: please, add a few words about the need for at least one 'live' contact between the hearing aid dispenser and the client/patient to monitor the 'technical' aspects and ensure that the provision and use of hearing aids is as comfortable and adequate as possible.

Line 316 ABR: same as for DPOAE’s: please add some words about the fact that remote ABR measurements must be done by trained personnel, so this differs from tele-audiometry, e.g. DIN-test, which can be performed by clients/patients themselves, using their smartphones incl. Apps and remote advice from hearing care professionals.

Line 330 Cochlear implants

Line/paragraph 338 remote CI assistants: please add some words to the difference between using Remote Assistant devices like CN CR120, which have to be used by medical experts in a remote situation, and tele-audiology through smartphones with App’s in general which can be used by patients/clients themselves, which is totally different from the example in the first case.

Line 372 it is not clear what a single on-site visit means: preoperative workup, surgery, and postoperative programming using remote assistant devices, or something else?

Line/paragraph 379 iCBT, PTM, Tele-PTM: please, pay some attention to the necessity that medical experts play an interactive remote role to follow and support the patient. If this is not the case, the therapy will not succeed! The way how this remote role has to be fulfilled has still to be investigated in near future.

Line 424 Tele-otoscopy, please add the fact the primary health care providers use this tool in remote connection with ENT-specialists to judge the otoscope images!

Line 447 otosope telemedicine via WeChat, please discuss the large difference between the outcome of studies [61] and [62], what is the reason for the large differences?

Line 451/paragraph Referral: please, add a little explanation how telemedicine specialty referral takes place, compared to standard primary care referral, because this not clear!

Line 461 standard

Line 504/paragraph Ethical Concerns: please, add some words about the problems to be tackled with the introduction of tele-audiology, as has been mentioned in different referenced papers, like privacy issues, how to organize who has rights to see measured online data, how to organize safety measures with respect to data transfer, etc.

Line 545/paragraph Licensure: a lot more about licensure and normalization procedures plays a role with the introduction of tele-audiology, as has been mentioned in different articles; almost nothing has been normalized until now, please add some words about what is missing, what is necessary, what has to be done in this respect.

Line 818 Ölcek, G.; Celik, I.; Başoğlu, Y.

General remark: references are marked with a number, sometimes accompanied by the name of first author, but not always, so, please adjust, always mention first author when the number of the reference is mentioned for the first time.

Comments on the Quality of English Language

No comments, other than minor editing of English language required.

Author Response

Dear reviewer,

Thank you.

Reviewer 2 Report

Comments and Suggestions for Authors

The authors reviewed the results and the impact of telemedicine in audiology. The discussion has include the application, the challenges and issues as well as the future advancement. It was an interesting review and may update the readers knowledge on telemedicine in the management of hearing impairment. In general, the article was well written.

Minor comments:

1. The date of the research started - 7 Oct 2024?

2. For certain examination, only the type of the test is mentioned followed by the results. It would be nicer if the authors may explain the techniques or the procedure of the test done remotely. 

Author Response

Dear reviewer,

Thank you very much.
